# Antifouling Property of Oppositely Charged Titania Nanosheet Assembled on Thin Film Composite Reverse Osmosis Membrane for Highly Concentrated Oily Saline Water Treatment

**DOI:** 10.3390/membranes10090237

**Published:** 2020-09-16

**Authors:** Nor Akalili Ahmad, Pei Sean Goh, Abdul Karim Zulhairun, Ahmad Fauzi Ismail

**Affiliations:** Advanced Membrane Technology Research Centre (AMTEC), School of Chemical and Energy Engineering, Faculty of Engineering, Universiti Teknologi Malaysia, Johor 81310, Malaysia; nor_akalili@yahoo.com (N.A.A.); zulhairun@petroleum.utm.my (A.K.Z.); afauzi@utm.my (A.F.I.)

**Keywords:** reverse osmosis, layer by layer assembly, thin film composite, oily saline wastewater, antifouling

## Abstract

With the blooming of oil and gas industries, oily saline wastewater treatment becomes a viable option to resolve the oily water disposal issue and to provide a source of water for beneficial use. Reverse osmosis (RO) has been touted as a promising technology for oily saline wastewater treatment. However, one great challenge of RO membrane is fouling phenomena, which is caused by the presence of hydrocarbon contents in the oily saline wastewater. This study focuses on the fabrication of antifouling RO membrane for accomplishing simultaneous separation of salt and oil. Thin film nanocomposite (TFN) RO membrane was formed by the layer by layer (LbL) assembly of positively charged TNS (pTNS) and negatively charged TNS (nTNS) on the surface of thin film composite (TFC) membrane. The unique features, rendered by hydrophilic TNS bilayer assembled on TFC membrane in the formation of a hydration layer to enhance the fouling resistance by high concentration oily saline water while maintaining the salt rejection, were discussed in this study. The characterization findings revealed that the surface properties of membrane were improved in terms of surface hydrophilicity, surface roughness, and polyamide(PA) cross-linking. The TFC RO membrane coated with 2-bilayer of TNS achieved >99% and >98% for oil and salt rejection, respectively. During the long-term study, the 2TNS-PA TFN membrane outperformed the pristine TFC membrane by exhibiting high permeability and much lower fouling propensity for low to high concentration of oily saline water concentration (1000 ppm, 5000 ppm and 10,000 ppm) over a 960 min operation. Meanwhile, the average permeability of uncoated TFC membrane could only be recovered by 95.7%, 89.1% and 82.9% for 1000 ppm, 5000 ppm and 10,000 ppm of the oily saline feedwater, respectively. The 2TNS-PA TFN membrane achieved almost 100% flux recovery for three cycles by hydraulic washing.

## 1. Introduction

Demand for fresh water has been accelerating exponentially with the overgrown population and rapid industrialization. Water reclamation from various potential sources has become an attractive solution to fulfil the increasing demand for freshwater [1]. Oily saline wastewater, also known as produced water (PW), is a potential water source as it is one of the largest waste streams produced in the oil and gas industry [2]. By eliminating the dissolved and suspended components present in PW, the treated water can be used to serve multiple purposes, including potable and non-potable uses. Water reclamation from PW also offers another advantage related to handling and disposal of hazardous PW. Oily saline wastewater containing a high amount of dissolved and dispersed hydrocarbons, surfactants, clay particles and salts is considered as one of the main water pollutants. The allowable limit for the total oil and grease discharged is normally set in the range of 10–15 mg/L [3,4]. Oily saline wastewater needs to be treated in order to effectively alleviate its adverse environmental impact. Currently, many approaches, including membrane technology, have been attempted for the treatment of oily saline wastewater [5,6,7].

Among the membrane processes, reverse osmosis (RO) has been touted as an effective process to treat oily saline wastewater generated from oil and gas industry [8,9]. This technology is matured and capable in removing many types of molecules, including oil particles and ions, to yield fresh water that is suitable for potable and industrial uses [5,10,11,12]. Fouling is one of the huge barriers to the widespread implementation of membrane-based filtration, particularly when treating saline water with high oil concentration. It is caused by the build-up of particulate and colloidal matter, such as oil droplets or proteins, that are deposited on the surface and/or within the pores of the membrane [13,14]. For PW desalination, the droplet coalescence of emulsified oil found in oily saline wastewater can easily foul the RO membranes. The stabilized emulsion PW is the most stable oil droplet with a size smaller than 10 μm and which tends to remain on the membrane surface [2]. Fouling by these oil droplets takes place within a short period and significantly shortens the operation lifetime of the RO membrane [15]. Therefore, to facilitate the application of the RO system for PW desalination, the membranes must be endowed with excellent anti-fouling properties to reduce the interaction between the foulants and membrane surface [16]. Membrane surface modification is one of the most versatile techniques to combat RO membrane fouling [17]. The introduction of surface modifiers, such as hydrophilic nanomaterials and antimicrobial polymers, renders excellent fouling resistance and enhanced membrane longevity and performance [18,19,20,21,22]. When dealing with hydrophobic foulant, such as emulsified oil, it is generally agreed that smooth and hydrophilic membrane surfaces can limit the surface–foulant hydrophobic interaction, hence suppressing membrane fouling [23].

PW treatment through a single RO process is preferable for reducing the cost and membrane downtime [24,25]. To achieve this sustainable approach, the RO membrane should be endowed with high fouling resistance [10,26]. The state-of-the-art membrane modification to cater for single stage RO oily saline wastewater treatment includes development of nanocomposite membranes and surface modification of commercially available membranes. Kasemset et al. performed surface modification of an RO membrane using polydopamine coating to improve water flux and fouling resistance [27]. During the single-stage RO PW treatment operation, notable water flux decline was observed due to the mass transfer resistance of the polydopamine (PDA) coating. Pei and co-workers developed a novel type of functional monomer hyperbranched polyamidoamine (PAMAM)/1,3,5-benzenetricarbonyl trichloride (TMC) RO membrane to form a thin and smooth TFC RO membrane to improve the antifouling of RO membrane [11]. Despite the well-maintained water flux for long hour operation, the salt rejection capability (89%) of the membrane was compromised. These works indicated that surface modification of membrane is a feasible approach to improve the performance of single-stage RO for PW treatment; however, a more holistic approach is needed to strike a good balance between the separation performance and fouling resistance. 

Layer by layer (LbL) assembly through the dip coating technique is an attractive technique for membrane surface modification [28,29,30]. The alternate stacking of anion and cation nanostructures, i.e., the number of bilayers can be carefully controlled during the deposition process on the polyamide (PA) layer to render desired surface properties [21,31,32,33,34]. In our recent work, LbL assembly of titania nanosheet (TNS) has been performed on PA RO membrane for PW desalination [22]. Compared to conventionally used titanium dioxide/titania nanoparticle (TiO_2_), TNS exhibits a large surface area and offers higher surface hydrophilicity owing to the presence of a large quantity hydroxyl (-OH) functional group [22,35]. A hydrophilic surface can mitigate the absorption of oil and induce the affinity of water to pass through membrane. Although the LbL surface coating could offer the additional resistance of water permeation, it also serves as a protective and sacrificial layer to the underlying PA layer. Apart from this, the transport characteristic of TNS carried a good, responsive membrane performance in terms of water permeability and salt rejection [24]. The 2D channels of TNS promise high water permeability. Additionally, the interaction between the OH polarity and the water molecules through the hydrogen bonds creates a favorable condition for fast water permeation [22].

Despite the successful surface modification of the RO membranes to improve flux while maintaining the salt rejection ability, the LbL TNS assembly also offers a great feasibility in designing an antifouling membrane long-term operation. Herein, this work is set to investigate the potential of LbL TNS assembly in combating fouling in the salt–oil–water separation process by using the optimal hydrophilic and fouling resistant TNS-PA TFN RO membrane. The performances were evaluated through dead-end RO filtration to separate highly concentrated oily saline water. The fouling-resistant mechanism of the TNS-PA TFN membrane was also investigated. 

## 2. Materials and Methods

### 2.1. Materials

Polymer pellet form (Solvay, Philadelphia, PA USA) of polysulfone (PSf, Udels P3500) was combined with polyvinylpyrrolidone (PVP, K30, Acros Organic, Morris, NJ, USA) and *N*-Methyl-2-pyrrolidone (NMP, 99.5%, Acros Organic, Morris, NJ, USA) to form the membrane support layer. Monomers used for the formation of PA active layers were m-phenylenediamine (MPD, 99.0%, Merck, Billerica, MA, USA) and TMC (98%, Acros Organic, Geel, Belgium) in n-hexane (C6H6, 49.0%, Merck, Billerica, MA, USA). Potassium carbonate ($K_{2}{\mathrm{CO}}_{3}$, 99.7%), lithium carbonate (${\mathrm{Li}}_{2}{\mathrm{CO}}_{3}$, 99.0%) from Vcem, titanium oxide powder (${\mathrm{TiO}}_{2}$, Degussa P25 Evonik), and hydrochloric acid (HCl, fuming 37%, ACL Labscan, Billerica, USA) were used to form positively charged titania nanosheets (pTNS) and by proceeding with tetrabutylammonium hydroxide (TBAOH, 40 wt.% in $H_{2}O$, Sigma-Aldrich, Louis, MO, USA) the negatively charged titania nanosheets (nTNS) were formed. The oily saline water consists of crude oil sample (Sabah Crude Oil Terminal, Sabah, Malaysia), sodium dodecyl sulfate (MW: 348.48 g/mol, SDS, Aldrich) as surfactant, and sodium chloride (NaCl, 99.5%, RCL Labscan, Bangkok, Thailand) and deionized water (DI) were mixed and used as feedwater throughout the RO performance studies.

### 2.2. Preparation of PA Composite Membrane with TNS Bilayers 

The preparation of pTNS and nTNS, as well as the preparation of TNS-coated PA TFC membrane, has been reported in our previous work [24]. In brief, the preformed neat TFC membrane was dipped into pTNS solution, followed by nTNS solution. The dipping was repeated to obtain the desired bilayer number. In this study, the TFC membrane was coated with 2 bilayers of pTNS-nTNS, i.e., 2TNS-PA TFN was used to investigate the antifouling performance in relation to the uncoated TFC. 

### 2.3. Characterization of Membrane

The hydrophilicity and oleophobicity of the membrane surface were determined from the dynamic water contact angle and oil contact angle measurements with DataPhysics contact angle (CA) 15Pro contact angle goniometer using Millipore RO water and crude oil, respectively, as probe liquid. The membrane CA was recorded at room condition with respect to the function of time. The contact angle (CA) system (708381-T, LMS Scientific, Filderstadt, Germany) was used to evaluate the water and oil contact angle measurement. The TFC and TFN membranes were cut off with a small rectangular piece of sample of 3 × 70 mm length and width and placed on the platform. The average reading from 5 measurements were reported. X-ray photoelectron spectroscopy was used to determine the elemental composition of composite membrane by using Kratos Axis Ultra DLD with aluminum Kα (1486 eV) radiation. From the atomic concentration of elements at the membrane surface, the degree of cross-linking of the PA layer can be obtained by Equation (1) by following previous published methods [36,37].
(1)$$\mathrm{Degree}\;\mathrm{of}\;\mathrm{cross}-\mathrm{linking}\left(\%\right)=\frac{m}{m+n}\times 100\%$$
where m is the cross-link of the PA layer and n is the linear parts of the PA layer. The values of m and n can be calculated based on the experimental O/N ratio obtained from XPS analysis using Equations (2) and (3).
(2)m+n=1
(3)$$\frac{O}{N}=\frac{3m+4n}{3m+2n}$$

Transmission electron microscopy (TEM) was used to confirm the successful coating of TNS onto the PA layer. Prior to TEM analysis, the formulation of epoxy resin formation (Eponate 12, Ted Pella. Inc., Redding, CA, USA) that was applied to embed the membrane was followed in the previous study [36]. The mixing of 0.8 g of 2,4,6-tri(dimethylaminomethyl)phenol, 14.3 g of nadic methyl anhydride, and 16.0 g of dodecenyl succinic anhydride and 29.0 g of resin was achieved by the medium hardness resin block. Then, liquid nitrogen was used to cool both the knife and specimen, which were adjusted down to −185 °C. After that, PT-PC Power Tome ultramictomes (RMC Boeckeler) w used to slice the resin block into thin sections with thickness of approximately 50 nm. The TEM images were then taken by using Hitachi HT770 under 120 kV. Surface morphology of the membrane of TFC and TFN membranes were visualized to study their antifouling property under scanning electron microscope (SEM; HITACHI TM3000). Zetasizer Nano (Litesizer 500 Anton Paar, St Albans, UK), with refractive indices of 1.5 and 1.333 for the oil particle and RO water, respectively, was utilized to measure the oil particle sizes of synthetic oily solution and permeate.

### 2.4. Evaluation of Membrane Performance by RO Testing Mode

#### 2.4.1. Water Permeability and Rejection Test of Saline Water

For the RO testing mode, a dead-end cell (HP4750, Sterlitech Corp, Kent, UK) with an effective surface area of 14.6 cm^2^ was used. Upon the placement of membrane into the dead-end cell, the membrane was compacted at 16 bar for 30 min to achieve steady water permeability under a constant magnetic stirring at 300 rpm. Then, for 15 min, the membrane was stabilized with RO water at 15 bar. Next, the permeated water was collected for 1 mL of RO water at 15 bar and the time taken was recorded. The filtration was repeated using the feed of 2000 ppm NaCl solution. The permeated water was collected for 1ml and the time taken was recorded. Water permeability of salt (A_w,s_) and salt rejections (R) were calculated using Equations (4) and (5):(4)$$A_{w,s\;}\left(L\cdot m^{-2}\cdot h^{-1}\cdot {\mathrm{bar}}^{-1}\right)=V/\mathrm{At}\Delta P$$
where V is the total amounts of the collected permeate (L), A is the cross-sectional area ($m^{2}$) and t is the time treatment (h).
(5)$$R_{s}\left(\%\right)=100\times \left(1-\frac{C_{p,s}}{C_{f,s}}\right)$$

$C_{f,s}$ is salt concentrations of feed and $C_{p,s}$ permeates the side measured by a conductivity meter.

#### 2.4.2. Water Permeability and Rejection Test of Synthetic Oily Saline Water

The synthetic oily saline solution was prepared by mixing the salt solution with crude oil [38]. The water was prepared by mixing crude oil in the range of 1000–10,000 ppm and SDS at the ratio of 9:1. A high speed blender (Model; BL 310AW, Khind, Shah Alam, Malaysia) was used to blend the solutions for 2 min with an agitation speed of 50 Hz at room temperature. Oil rejection was measured by using a UV–vis spectrometer (DR2800, Loveland, CO, USA). The absorbance from UV–vis was measured at 281 nm and the relation between the absorbance and oily saline feedwater found to be linear as shown in Appendix A.

The water permeability of oily saline (A_w,o_) and oil rejection (R_o_) was calculated using Equations (6) and (7):(6)$$A_{w,o\;}\left(L\cdot m^{-2}\cdot h^{-1}\cdot {\mathrm{bar}}^{-1}\right)=V/\mathrm{At}\Delta P$$
where V is the total amounts of the collected permeate (L), A is the cross-sectional area ($m^{2}$) and t is the time treatment (h).
(7)$$R_{O}\left(\%\right)=100\times \left(1-\frac{C_{p,o}}{C_{f,o}}\right)$$
where $C_{p,o}$ and $C_{f,o}$ represent the concentration of oil permeate and oil feed (abs), respectively.

#### 2.4.3. Antifouling Test

The antifouling test was conducted in RO mode by using a various concentration of oil emulsion (1000 ppm, 5000 ppm and 10,000 ppm of crude oil) in saline water (2000 ppm NaCl) as a feedwater. The filtration was performed for 960 min in four cycles in order to observe the permeability change as a function of time. The permeate was collected and recorded at the interval of 30 min. After each test, the membrane was washed by distilled water for 30 min before proceeding with another cycle. Water permeability of the coated TFC and uncoated TFN membranes of each membrane was periodically measured to investigate the fouling behavior of the membrane.

The irreversible fouling ratio (R_ir_) was calculated based on the following Equation (8):(8)$$R_{\mathrm{ir}}\left(\%\right)=100\times \left(1-\frac{J_{\mathrm{wp}}}{J_{\mathrm{wv}}}\right)$$
where $J_{\mathrm{wp}}$ and $J_{\mathrm{wv}}$ represent the flux of fresh membrane and washed membrane, respectively.

## 3. Results and Discussion

### 3.1. Effect of TNS Loading on Membrane Chemistry

#### 3.1.1. Surface Hydrophilicity of TNS-PA TFN Membranes

Surface hydrophilicity is an important factor to improve the membrane performance for oily saline water separation. Figure 1a displays the dynamic water contact angles of TFC membrane assembled with different TNS bilayer as the function of time. The surface contact angles of TFC membrane reduced from 47.9° to 36.4° within 660 s. Compared to the uncoated TFC membranes, all TNS coated PA membranes exhibited more significant decline in the water angle. The highest decline was achieved by the 4TNS-PA TFN membrane in which the water contact angle reduced from 45.32° to 21.61°. It was observed that the initial water contact angle of the TFC membrane was slightly lower than the coated membrane due to the wetting properties of the rough TFC membrane surface caused by Wenzel and Cassie effect [39]. Based on the water contact angle decline pattern, it can be deduced that the TNS coated membrane was enriched with a higher amount of surface hydrophilic groups. 

Underwater oleophobicity was evaluated by measuring the underwater oil contact angle as shown in Figure 1b. The underwater oil contact angle of pristine TFC membrane was measured as 105°, while all coated TNS-PA TFN membranes exhibited an oil contact angle of >135°. The result indicated that the hydrophilicity of TNS has effectively reduced the interaction with the hydrophobic oil droplets, due to the formation of the surface water layer. It is known that the formation of the hydration layer is associated with better membrane oleophobicity where the high intensity of the hydration layer resulted in a higher oil contact angle [40]. During the filtration process, this favorable layer could prevent the oil from wetting the surface underwater and provides a better cleaning function.

As can be seen in Figure 1c, during the underwater oil contact angle measurement, all modified membranes experienced a low tendency of oil droplet wetting and were less affected by the adhesion force with the membrane surface. The oil droplet that dripped on the membrane surface experienced the “attach-compress-detach” patterns for several times. Moreover, there was no obvious deformation of oil droplets observed when the tip with oil droplet was released (force released) from the membrane surface. This can be attributed to the presence of a layer of thick water cushion. 

#### 3.1.2. Morphology and Chemical Compositions of TNS-PA TFN Membranes

Based on our previous work, the 2TNS-PA TFN membrane was selected as the optimal membrane among the coated membranes due to its permeability and salt rejection performance [22]. To further verify the coating of oppositely charged TNS, the morphology and surface element content were evaluated from the cross-sectional TEM image and XPS full-scan spectra, respectively, as shown in Figure 2a,b. Ridge and valley structure of PA layer formed from the in interfacial polymerization (IP) technique can be clearly observed from the TEM image in Figure 2a. This PA layer structure was covered by TNS layers that could be clearly seen within the PA layer. TNS slightly penetrated into the PA layer due to the presence of void and space within the ridge and valley structure. The penetration of TNS reduced the thickness of this coating layer, which could otherwise increase the mass transport resistance. As stated by Rajaeian et al., the settlements of nanoparticle throughout the PA matrix could help the formation of dense and smooth morphology of the coated membrane [41].

Figure 2b shows the high resolution XPS full-scan spectra of pristine TFC and TFN with optimal coating of TNS (2-bilayer). Based on the XPS depth profiling analysis, X-ray beam can penetrate the membrane sample in the range of 5–10 nm from the membrane surface [36,42,43,44]. The peaks at approximately 284, 397 and 531 eV were attributed to O 1s, N 1s and C 1s, respectively, implying the formation of PA layer. Upon the surface modification with TNS on the PA layer, the photoelectron peak of Ti atom (Figure 2b) appears at a binding energy, Eb = 456.2 eV for Ti 2p3/2 and 461.9 eV for Ti 2p1/2, as well as for the O atom at 531.0 eV. Additionally, a small percent of nitrogen was detected from the PA layer of the 2TNS-PA membrane as it was within the penetration depth of the XPS X-ray beam. These peaks suggested that the TNS layers have been successfully coated on the PA membrane surface. It is obvious that the concentration of element O from the 2TNS-PA TFN membrane is also much higher than the TFC membrane. The dramatic increase in the O element content suggested that the 2TNS-PA TFN membrane surface has a more exposed hydrophilic polar group of -OH than the TFC membrane. Ti-OH bonds on the outer membrane surface that absorbed the -OH group of the water molecules could facilitate the transport of water molecules through the membrane, thereby increasing membrane permeability. Thus, the results indicated that TNS has sufficient binding strength with the PA membrane. 

To further support the mechanism of enhanced permeability, the cross-linking degree of the PA layer for both the uncoated and coated membranes was obtained from the XPS analysis. Table 1 shows the elemental compositions, the relative ratios of O/N and C/N, and the degree of the cross-linking of the PA active layer for both TFC and 2TNS-PA TFN membranes. The element ratios of O/N reflect the layer cross-linking degree of PA layer. Theoretically, the ratio falls between 1.0 and 2.0 A value of 1.0 indicates that the PA layer is fully cross-linked, while a value of 2.0 corresponds to the fully linear structure [45]. The relative atomic concentration of C, O, N, and Ti as well as corresponding C/N and O/N ratios is tabulated in Table 1. Table 2 indicates that the O/N ratio of the 2TNS-PA TFN membrane was higher than the TFC membrane, indicating the lower degree of amide cross-linking (-NH-CO-) in the former. While the rest of the percentage refers to the linear pendant carboxylic acid groups (-COOH) in the PA membrane. The lower degree of PA cross-linking was due to the penetration of TNS on and within the PA layer as evidenced from the TEM images (Figure 2a). The LbL assembly of TNS has favorably disturbed the PA layer by loosening the network structure as evidenced by the reduction in cross-linking. In addition, this disturbance may increase the oxygen content when the portions of acid halide groups of TMC not fully cross-linked with the amine monomer and unreacted acid chloride groups would be hydrolyzed to -COOH. These possibilities were in good agreement with Sagle et al., who modified the surface of the PA RO membrane with polyethylene glycol [44]. 

### 3.2. Separation Performance of Membranes

Figure 3 shows the separation performances based on the permeability and rejection of five membranes including the uncoated TFC and coated membranes tested for oily saline water separation. In Figure 3a, the TFC membrane demonstrated water permeability of 0.65 L·m^−2^ h^−1^ bar^−1^ and slightly increased for the 1TNS-PA TFN membrane with 0.71 L·m^−2^ h^−1^ bar^−1^. The TFC membrane assembled with 2-bilayer of TNS has a higher water permeability with 0.98 L·m^−2^ h^−1^ bar^−1^. However, with the increasing bilayer deposition up to ≥3 bilayers, the pure water and water permeability decreased due to the unfavorable thickness [46]. The additional mass transfer resistance created on the surface of the RO membrane offsets any hydrophilic benefit of the coating [22].

When oily saline water with lower surface tension and oil–water droplets was used as feedwater, all tested membranes experienced a significant reduction in permeability. Although a similar decline has been observed for all the membranes, the 2TNS-PA TFN membrane permeability exhibited the highest permeability. This was attributed to the balanced hydrophilicity improvement and surface coating thickness obtained by the 2TNS-PA TFN membrane. The high surface hydrophilicity preferentially attracted water rather than oil, hence increasing the higher permeability [22]. The membrane was endowed with high water capture capacity with the formation of a hydration water layer near the surface. Consequently, the contact of oil droplets with the membrane surface could be suppressed and the stable underwater oleophobicity of the membrane was maintained. This observation shows the enhancements of fouling resistance of hydrophilic membrane modification as consistent with the dynamic oil contact angle result. 

Besides that, the low PA cross-linking degree of 2TNS-PA TFN membrane also contributed to the increase in water permeability. The loosened PA matrix from the top membrane formed the additional water transport channel and allowed a greater water uptake. The stable and strong H-bonding formation between water and membrane surface would promise more selective permeation. The higher stability of the H-bond of the -COOH group with water would allow high water permeability. As stated by Zhang et al., the -COOH groups in the PA layer formed a more stable and strong H bond with water than hydrophilic amide groups [47]. Thus, the water capacity initiated a faster flow of water molecules to pass through to the rest of the PA membrane. The PA layer that consists of cross-linked (-NH-CO-) linkage and linear pendant (-COOH) facilitated the diffusion of water molecules through the PA thin film and elevated the water permeability. Moreover, as based on the TEM images, an adequate TNS bilayer coating causes the coated membrane thickness almost similar with the uncoated membrane. It can be concluded that the performance of 2TNS-PA TFN membrane did not affect the membrane mass transport and could achieve high permeability. 

As presented in Figure 3b, all the tested membranes exhibited salt and oil rejections of ≥98% and ≥99%, respectively. High oil rejection was expected based on the dense structure of both coated and uncoated PA membranes. This could be described by the fact that the all membranes were characterized by a dense structure that is able to effectively hinder the diffusion of the oil–water emulsion through the membrane. Despite the dense structure, the PA selective layer may also suffer from minor defects [48]. Upon the hydrophilic layer coating of TNS, the minor defects of the PA layer can be sealed. Sagle et al. observed the increase in solute rejection which can be attributed to the hydrophilic layer coating [44]. The oil–water emulsion layer also played a role in affecting the rejection. With the addition of anionic SDS into the oil–water of feedwater, the ionic strength will increase and lead to less droplet–droplet repulsion [49,50,51]. Thus, the oil droplet formed a dense cake layer. Consequently, the concentrated layer of oily saline emulsion reduced the permeability of oily saline water while increasing the rejection. In the subsequent sections, the 2TNS-PA TFN membrane with the most satisfying permeability and rejection performance was selected for the antifouling studies.

### 3.3. Antifouling Behaviour of Membranes

Oil emulsion fouling is a great challenge in membrane-based separation of wastewater variety including oily saline water from oil and gas recovery. A prolonged filtration study was performed to demonstrate the antifouling properties of the 2TNS-PA TFN membrane in relation to the TFC membrane. Figure 4a–c presents the fouling behavior of TFC and 2TNS-PA TFN membranes with feedwater concentrations of 1000 ppm, 5000 ppm and 10,000 ppm oily in constant saline water concentration of 2000 ppm over 960 min in total of four cycles. During the prolonged filtration process, both membranes were cleaned for every cycle by hydraulic flushing.

In Figure 4a–c, both TFC and 2TNS-PA TFN membranes experienced fouling as expected. The average permeability decline was calculated to characterize the fouling phenomena. Under feed concentrations of 1000 ppm and 5000 ppm, oily saline feedwater—as shown in Figure 4a,b—TFC membrane demonstrated significant permeability drops, which were 38.8% and 41.8%, respectively. The decline was further increased to 52% when the TFC was operated under the high feedwater concentration of 10,000 ppm (Figure 4c). The surface of uncoated the TFC membrane was prone to the deposition of oil particles as it possessed higher roughness compared to the TNS-coated membrane [24]. The oil particles could be easily trapped within the rougher surface. In addition, a cross-linked portion of amide linkages and a linear portion of carboxylic groups of pristine TFC membrane could facilitate the H bonds between the -OH groups around the emulsified oil particles and the TFC layer [52]. Moreover, the oil droplet size affects the severity of the TFC membrane fouling due to the larger surface area and stronger attraction force with membrane surface [52]. As a result, the attachment of oil particles onto the TFC surface unfavorably took place. Therefore, the TFC surface was easily fouled by the emulsified oil particles. 

Meanwhile, as can be seen in Figure 4a,b, the average permeability decline of 2TNS-PA TFN membrane was 31.7% and 35.0% for oily saline feedwater concentrations of 1000 ppm and 5000 ppm, respectively. On the other hand, the 2TNS-PA TFN membrane could maintain its average permeability drop with 44.0% for the high oily saline feedwater of 10,000 ppm (Figure 4c). In comparison with the TFC membrane, the 2TNS-PA TFN membrane demonstrated better permeability stability in all three feed concentrations. The high hydrophilicity, smooth surface and low PA cross-linking of 2TNS-PA TFN membrane have concertedly contributed to better antifouling properties. The TNS layer could protect the outermost layer of membrane from the attachment of the emulsified oil particles. Besides that, the smooth membrane surface due to surface coating also rendered a better fouling resistance by ensuring a small contact area and weak interaction between the membranes and oil droplets [10]. Moreover, the low cross-linked PA layer was due to the presence of TNS layer on top, which reduced the amount of emulsified oil particles approaching the TFC layer. It serves as a protective and sacrificial layer to the underlying PA layer. Thus, this newly developed TNS-PA TFN membrane can overcome the fast fouling problem of the TFC membrane’s oily saline water separation. As far, the TNS on PA layer possesses remarkable delayed fouling and higher performance recovery upon membrane hydraulic cleaning would be favorable for the efficient application in oil/water separation. 

The fouling of TFC and 2TNS-PA TFN membranes was quantified by calculating the Rir and the deposition of foulants was visualized using SEM imaging. Figure 5 presents the SEM images of the fresh, fouled and water-cleaned TFC and 2TNS-PA TFN membranes. Compared to the fresh TFC membrane, the surfaces of both fouled and physically cleaned TFC membranes were deposited with visible foulants. This observation supports the earlier discussion that stated that severe fouling took place on TFC and the foulant cannot be easily removed by physical cleaning. As a confirmation, the obtained value of its irreversible fouling ratio (R_ir_) was high with 17.09%. On the other hand, TNS coating was clearly observed on the fresh TFN membrane. Compared with the TFC membrane, significantly less foulant was found deposited on the membrane surface. Upon cleaning, the foulants were removed and the original surface morphology was restored. This observation is in good agreement with the lower R_ir_ of 6.12%.

Appendix A indicates the physical appearance of the permeations after 960 min for different concentrations of oily saline water as feed (1000 ppm, 5000 ppm, 10,000 ppm). From the visualization, it could be observed that all the milky yellow-colored feedwater turned to transparent with no visible oil droplets. A similar observation has also been made by Qin et al. [52]. Zeta sizer Nano was used to confirm the presence of oil droplets and their sizes in all the permeates of different oil concentration filtered by the 2TNS-PA TFN membrane. The presence of oil molecules in the permeate can be detected when the light (scattered in all directions; Rayleigh scattering) hits small particles that undergo Brownian motion [53,54]. Notably, the intensity of the scattered light as a function of time refers to the size of particles. As shown in Figure 6a–c, the oil droplet size distribution in oily saline feedwater widened and the average oil droplet size increased when the oil concentration increased from 1000 ppm to 5000 ppm and 10,000 ppm. In an emulsifier-stabilized system, the oil droplets are covered by the emulsifier, SDS in this case, to maintain the small droplet size. However, with the increasing oil concentration in the oily saline feedwater, the amount of emulsifier was not sufficient to provide full coverage on the oil droplets, resulting in destabilization and partial coalescence of the oil droplets. Consequently, the average oil droplet particle size was increased [38]. Apart from this reason, the oil droplet size distribution is also known to be influenced by the presence of NaCl. Faster aggregation and coalescence take place at a sufficiently high ionic strength, hence increasing the droplets size [55].

By comparing the size distribution of permeate and feedwater, the distribution of oil droplet size after the separation, i.e., those collected from the permeate, was narrowed and characterized by smaller droplet size. This change was mainly contributed by the effect of intense shear force and droplet impact. Under constant magnetic stirring at 300 rpm during the prolonged testing, the oil droplet experienced a disruption force that resulted in droplet breakup [55]. Therefore, over the course of operation, the diffusion of the oil–water emulsion through the membrane may take place, as detected in all permeates collected. The detection of oil droplets from the permeate is in line with the incomplete oil rejection of the 2TNS-PA TFN membrane. 

The salt rejection and oil repellent properties of the 2TNS-PA TFN membrane are schematically presented in Figure 7. This separation mechanism was attributed to the hydration layer that intrinsically formed on a highly hydrophilic membrane surface with ultralow adhesion to the dispersed oil droplets. The water trapped in the membrane structure formed a continuous phase and provided a strong oil repelling force. The strong bound water molecules that created an energetic barrier occlude the tiny oil particles from binding to surface of membrane [56]. The water droplets from the salt–oil emulsion permeate to the opposite side of the coated TFN membrane through the channel of the absorbed water layer.

Table 3 tabulates and compares the performances of the membrane prepared in the present work with other membranes prepared with similar methods in treating oily saline wastewater. Overall, the treatment of oily saline wastewater using single-stage RO has not been extensively performed, mainly due to the challenges in dealing with high oil concentration in saline water. Ji et al. used a commercial TFC RO membrane to treat PW with low oil concentration and were able to obtain a high water flux and rejection of salt and oil [26]. Kasemset et al. coated commercial TFC RO membrane with polydopamine to have a better membrane performance in terms of water flux and rejection [27]. However, severe water flux decline was observed due to membrane fouling. In this study, the surface modification of RO membrane with TNS bilayer coating has resulted in an increase in water flux and antifouling property compared to TFC membrane. Compared to similar work, such as the one performed by Pei et al., the membrane developed in this study was able to sustain satisfactory rejection and flux, while demonstrating significantly improved antifouling properties. The potential of a two-dimensional TNS bilayer assembled on TFC membrane has been explored and shows better performance in demulsification and desalination. Despite the achievement, there is still plenty of room for improvement and one of them is to further heighten the water flux by refining the LbL assembly of the TNS bilayer.

## 4. Conclusions

In this study, the coating of TNS onto the PA membrane through the LbL technique demonstrated great feasibility for the separation of oil and salt from oily saline water through a single filtration step. The dynamic water contact angle of the TNS-coated PA TFN membrane suggested a significant increase in surface hydrophilicity, as compared to the pristine TFC. Besides that, the TNS coating reduced the degree of amide cross-linkage in PA layer, hence facilitating the water transport due to the higher stability of H-bond of -COOH group of PA layer with water. Combining these conditions, the 2TNS-PA TFN membrane demonstrated optimal performance in terms of water permeability and the simultaneous rejection of oil and salt. In the long-term antifouling test, both 2TNS-PA TFN and TFC membranes experienced expected permeability decline. However, the 2TNS-PA TFN membrane suffered a less significant decline and exhibited a relatively higher average permeability recovery rate compared to the TFC membrane due to the better resistance to the oil adsorption on the skin layer. Overall, the surface modification of TNS on the TFC membrane can provide stable and effective therapies to tackle the fouling problem in TFC membrane. The newly developed TFN membrane can be beneficially used for clean water production.

## Figures and Tables

**Figure 1 membranes-10-00237-f001:**
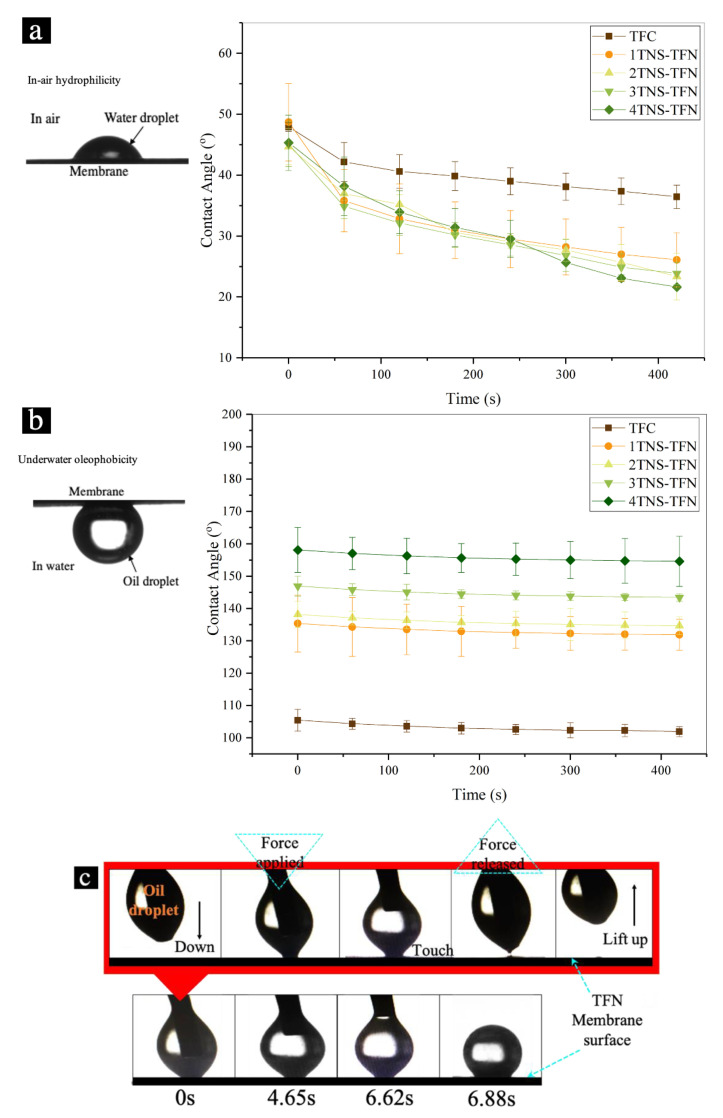
(**a**) Water contact angle with (**b**) underwater oil contact angle of all experimental membranes and (**c**) an “attach-compress-detach” behavioral pattern of oil droplets during contact with all prepared membranes’ surfaces before the values of the underwater oil contact angle can be taken.

**Figure 2 membranes-10-00237-f002:**
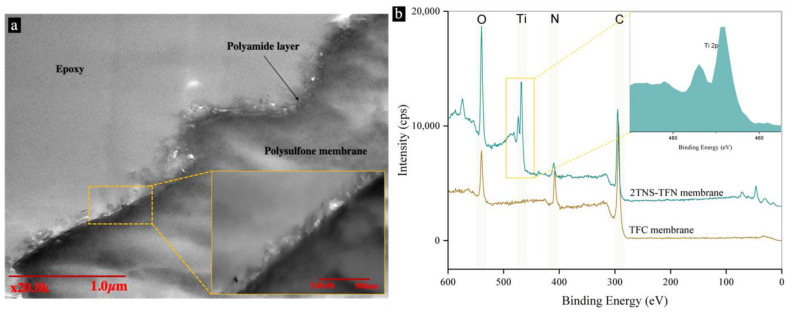
(**a**) TEM microscopic of 2TNS-PA TFN membrane with scale bar (inset: magnified TEM microscopic of PA layer) and (**b**) XPS full-scan spectra of TFC and 2TNS-PA TFN membranes. (Inset: the magnified peaks of TNS assembled on the surface of PA membrane.).

**Figure 3 membranes-10-00237-f003:**
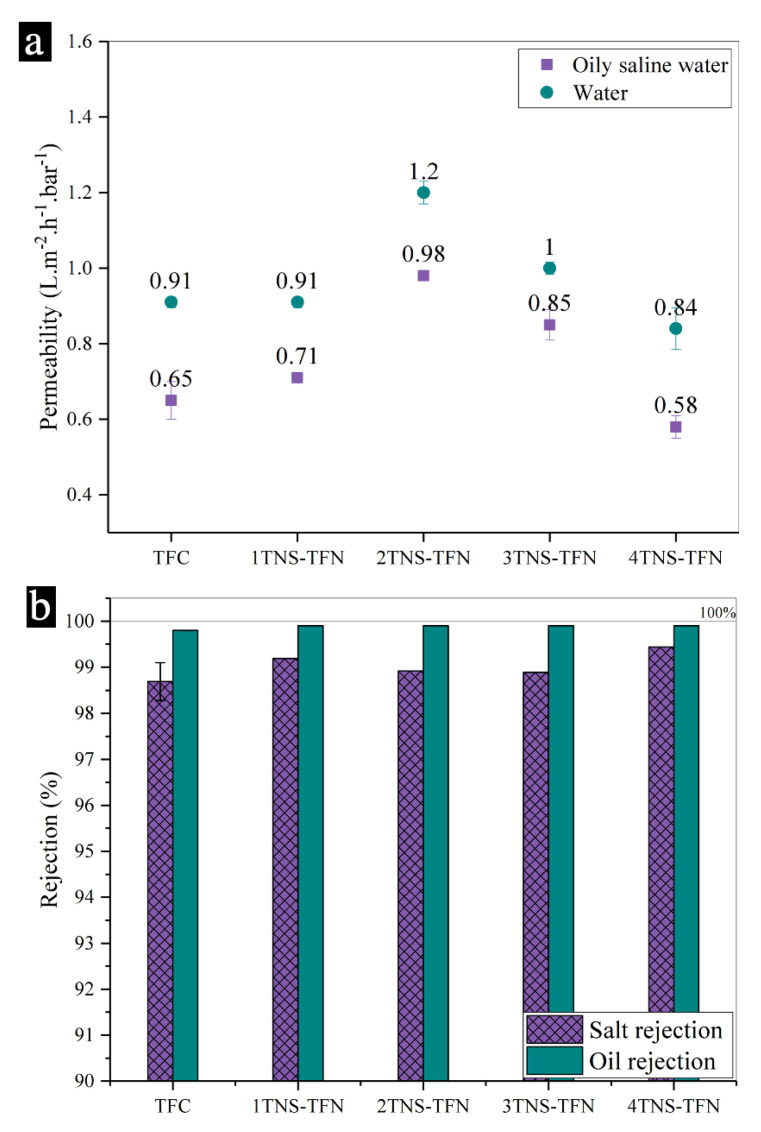
(**a**) Pure water and water permeabilities of membranes and (**b**) oil and salt rejections of composite membranes.

**Figure 4 membranes-10-00237-f004:**
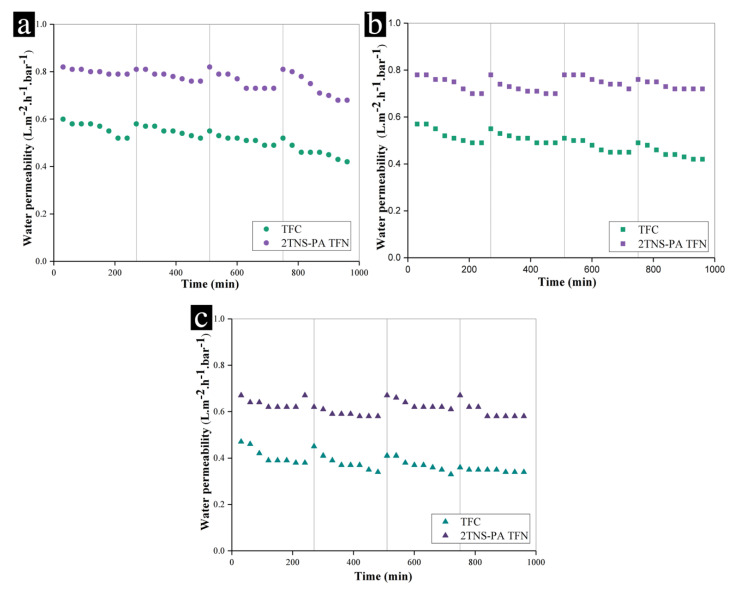
Comparison of water permeability for TFC and 2TNS-PA TFN membranes at high different oily saline feedwater of (**a**) 1000 ppm oily saline water, (**b**) 5000 ppm oily saline water and (**c**) 10,000 ppm oily saline water (grey lines represent the water recovery after washing).

**Figure 5 membranes-10-00237-f005:**
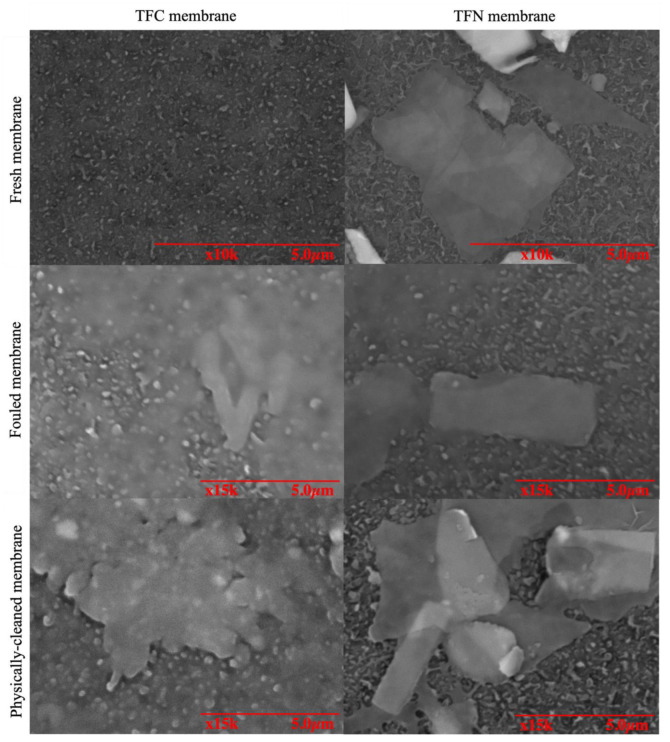
SEM images of TFC and TFN membranes of fresh, fouled and physically cleaned membranes.

**Figure 6 membranes-10-00237-f006:**
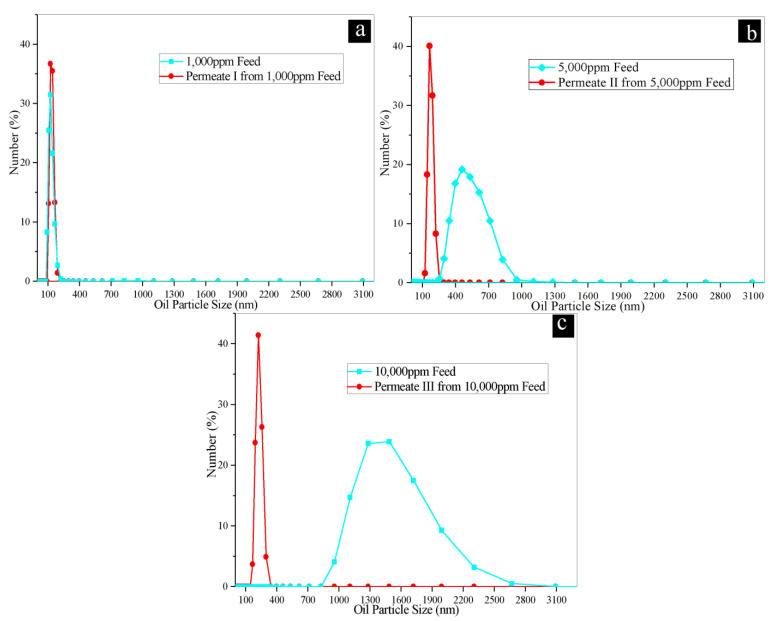
The oil particles (size distribution by number) of the synthetic oily saline wastewater at (**a**) 1000 ppm, (**b**) 5000 ppm and (**c**) 10,000 ppm as feedwater and the permeation of each concentration by the 2TNS-TFN membrane.

**Figure 7 membranes-10-00237-f007:**
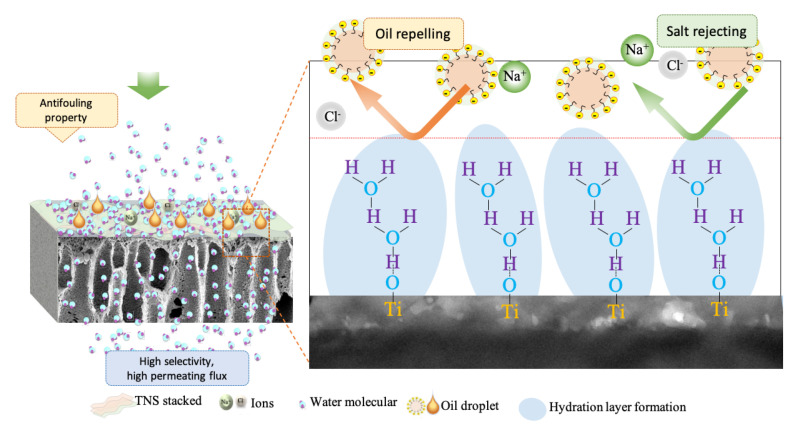
An explanatory sketch of oil repellent and salt rejection.

**Table 1 membranes-10-00237-t001:** Element composition, O/N ratio, C/N ratio and degree of cross-linking of TFC and TFN membranes.

Membrane	Atomic Concentration (%)				O/N Ratio	O/N Ratio	Degree of Cross-Linking (%)
	C (1s)	O (1s)	N (1s)	Ti (2p)			
TFC	75.35	13.39	11.26	-	1.19	6.69	74.10
2TNS-PA TFN	56.00	30.03	5.87	8.10	5.12	9.54	33.53

**Table 2 membranes-10-00237-t002:** Chemical structure of the repeating unit aromatic PA.

Membrane Type	Cross-Linked Portion of Amide Linkages/m (%)	Linear Portion of Carboxylic Groups/n (%)
TFC	74.10	25.90
2TNS-PA TFN	33.53	66.47

**Table 3 membranes-10-00237-t003:** Separation performance of reverse osmosis (RO) membranes towards produced water (PW).

Membrane	Flux	Rejection	Long Term Performance		Ref.
			Flux Drop	Flux Recovery	
Commercial RO (RE-4040) Feed: 4800–5500 ppm of salts 9–13 ppm petrolic	50–55 L·m^−2^·h^−1^	~99.9% of total organic carbon (TOC) 96.6% of salt	-After 15 h, flux declined 75% of its initial flux	-Recovery 100% (restored to its original state)	[26]
PSf-PAMAM+TMC Feed: 2.5 or 5 mL hexadecane 2000 ppm salt 250 mg SDS	18.42 L·m^−2^·h^−1^	99% of TOC 89.3% of salt	-After 18 h, flux declined 5% -After 24 h, flux declined 30%		[10]
Commercial RO incorporated with polydopamine Feed: 1500 ppm of soybean oil soybean oil 2000 ppm salt	>40 L·m^−2^·h^−1^	>99.1% of salt	-After 1 h, flux improved 30–50% from TFC	-Recovery 100% (restored to its original state)	[27]
PSf-MPD+TMC incorporated with TNS bilayers Feed: 1000 ppm oil 2000 ppm of salt	14.7 L·m^−2^·h^−1^	>99% of oil >99% of salt	-After 4 h, flux declined 5% -After 16 h, flux declined 17%	-Recovery 100% (restored to its original state)	This study

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
