# Peer review of "Antifouling Property of Oppositely Charged Titania Nanosheet Assembled on Thin Film Composite Reverse Osmosis Membrane for Highly Concentrated Oily Saline Water Treatment"

_membranes, 2020, doi:10.3390/membranes10090237_

Round 1

Reviewer 1 Report

An antifouling RO membrane for oily saline water treatment was prepared in this study. The topic is attractive. The experiments have been properly designed and the manuscript has been well prepared. Therefore, I suggest a revision before it could be accepted. Some detailed comments are as follows:

1# The introduction is too wordy. Please simplify paragraphs 1-5.

2# Carefully check the typo and use of grammar in the entire manuscript.

3# How does the antifouling RO membrane prepared in this study when compared to other membranes prepared with similar method.

Author Response

Reviewer 1

 An antifouling RO membrane for oily saline water treatment was prepared in this study. The topic is attractive. The experiments have been properly designed and the manuscript has been well prepared. Therefore, I suggest a revision before it could be accepted. Some detailed comments are as follows:

Thank you for the comments. All of the changes made are marked in the main text with green colour.

1# The introduction is too wordy. Please simplify paragraphs 1-5.

The introduction has been shortened. Please refer to section 1.0

2# Carefully check the typo and use of grammar in the entire manuscript.

The typo and grammar have been improved in the entire manuscript.

3# How does the antifouling RO membrane prepared in this study when compared to other membranes prepared with similar method.

The membrane prepared in this study is benchmarked with  other membranes prepared with similar method. Please refer to Table 3. The discussion is provided in Page 16 of the revised manuscript.

Reviewer 2 Report

Major Revision

The author has demonstrated the feasibility of preparation of antifouling RO membrane for oily saline water treatment. This is a good research topic that may arise few researcher's interests studying reverse osmosis and membrane fouling. However, in this research paper, there are few important issues that have been mentioned below. Therefore, I recommend major revision for enhancing the quality of this manuscript.

  1. Even though abstract consists of objectives and results but at the same time the author should highlight the novelty statement for attracting readers.
  2. Coming to next point, introduction seems to be very generalized. Lot of research works were already executed based on RO performance. The state-of-art is totally missing from the manuscript. The author should have added the previous research outputs (conditions/parameter/overall results) by comparing the present one in tabular form to show the viability of the present study.
  3. Section 2.4.1 and section 2.4.2 is same. Kindly revise the present manuscript thoroughly.
  4. A new section is needed to mention the characterization part in materials and methods section.
  5. What we can gain from Figure 2? There is no change in the FTIR. Kindly come up with high scientific discussion.
  6. Figure 3 must be replaced with high quality figure. All the figures must be improved in the revised manuscript especially graphical figures.
  7. While talking about the membrane fouling or scaling, the author must indicate following results which is totally missing from the manuscript (if possible):
  • SEM for morphological study (fresh and used membrane)
  • EDS for elemental composition (fresh and used membrane)
  1. Kindly cite 5-6 more articles from “Membranes”, “Polymer” MPDI journal in the introduction section.

Author Response

Reviewer 2

Major Revision

The author has demonstrated the feasibility of preparation of antifouling RO membrane for oily saline water treatment. This is a good research topic that may arise few researcher's interests studying reverse osmosis and membrane fouling. However, in this research paper, there are few important issues that have been mentioned below. Therefore, I recommend major revision for enhancing the quality of this manuscript.

Thank you for the comments. All of the changes made are marked in the main text with green and red colour.

  1. Even though abstract consists of objectives and results but at the same time the author should highlight the novelty statement for attracting readers.

The abstract has been improved by providing some novelty statement of this study. Please refer to line 16.

“This study focuses on the fabrication of antifouling RO membrane for accomplishing simultaneous separation of salt and oil. Thin film nanocomposite (TFN) RO membrane was formed by the layer by layer (LbL) assembly of positively charged TNS (pTNS) and negatively charged TNS (nTNS) on the surface of thin film composite (TFC) membrane. The unique features rendered by hydrophilic TNS bilayer assembled on TFC membrane in the formation of hydration layer to enhance the fouling resistance by high concentration oily saline water while maintaining the salt rejection were discussed in this study.”

  1. Coming to next point, introduction seems to be very generalized. Lot of research works were already executed based on RO performance. The state-of-art is totally missing from the manuscript. The author should have added the previous research outputs (conditions/parameter/overall results) by comparing the present one in tabular form to show the viability of the present study.

The state-of-the-art of oily saline wastewater treatment using RO is included in this manuscript. Please refer line 70-84.

The benchmarking of the membrane prepared in present work with other membranes prepared with similar methods is made in the revised manuscript. Please refer to Table 3 (page 16). The discussion is provided in Page 15 of the revised manuscript.

  1. Section 2.4.1 and section 2.4.2 is same. Kindly revise the present manuscript thoroughly.

Section 2.4.1 and 2.4.2 have been revised. Section 2.4.1 describes the water permeability and rejection test of saline water (at line 160) while Section 2.4.2 (at line 174) describes the water permeability and rejection of synthetic oily saline water. The sub-titles are revised and the symbols of equations are modified to avoid confusion.

  1. A new section is needed to mention the characterization part in materials and methods section.

A new section is added: “2.3 Characterization of membrane”.

  1. What we can gain from Figure 2? There is no change in the FTIR. Kindly come up with high scientific discussion.

Figure 2 and the discussionhave been omitted in the revised manuscript.

  1. Figure 3 must be replaced with high quality figure. All the figures must be improved in the revised manuscript especially graphical figures.

The quality of figures have been improved throughout the manuscript.

  1. While talking about the membrane fouling or scaling, the author must indicate following results which is totally missing from the manuscript (if possible):
  • SEM for morphological study (fresh and used membrane)
  • EDS for elemental composition (fresh and used membrane)

The SEM images of TFC and 2TNS-PA TFN membranes have been included in Figure 5, page 13 in the main text. Rir was calculated to quantify and compare the severity of irreversible fouling took place in the membranes. The discussion is provided in page 12, line 366.

  1. Kindly cite 5-6 more articles from “Membranes”, “Polymer” MPDI journal in the introduction section.

The articles from “Membranes” and “Polymer” MDPI journal have been added in the introduction section as following:

  1. Liu, L.F.; Gu, X.L.; Qi, S.R.; Xie, X.; Li, R.H.; Li, K.; Yu, C.Y.; Gao, C.J. Modification of Polyamide-urethane (PAUt) thin film composite membrane for improving the reverse osmosis performance. Polymers2018, 10, doi:10.3390/polym10040346.
  2. Elhady, S.; Bassyouni, M.; Mansour, R.A.; Elzahar, M.H.; Abdel-Hamid, S.; Elhenawy, Y.; Saleh, M.Y. Oily wastewater treatment using polyamide thin film composite membrane technology. Membranes2020, 10, 1–17, doi:10.3390/membranes10050084.
  3. Ostarcevic, E.R.; Jacangelo, J.; Gray, S.R.; Cran, M.J. Current and emerging techniques for high-pressure membrane integrity testing. Membranes2018, 8, 1–27, doi:10.3390/membranes8030060.
  4. Baransi-Karkaby, K.; Bass, M.; Freger, V. In situ modification of reverse osmosis membrane elements for enhanced removal of multiple micropollutants. Membranes2019,9, doi:10.3390/membranes9020028.
  5. Mayyahi, A. Al Important Approaches to Enhance Reverse Osmosis ( RO ) Thin Film Composite ( TFC ) Membranes Performance. Membranes2018, 8, 68, doi:10.3390/membranes8030068.
  6. Yang, Z.; Zhou, Y.; Feng, Z.; Rui, X.; Zhang, T.; Zhang, Z. A Review on Reverse Osmosis and Nanofiltration Membranes for Water Purification. Polymers2019, 11, 1252, doi:10.3390/polym11081252.

Round 2

Reviewer 2 Report

The authors have responded well with high scientific discussions. Therefore, the revised manuscript can be accepted in the present form.